# Just Pick a Sign: Optimizing Deep Multitask Models with Gradient Sign Dropout

**Zhao Chen**
Waymo LLC
Mountain View, CA 94043
`zhaoch@waymo.com`

**Jiquan Ngiam**
Google Research
Mountain View, CA 94043
`jngiam@google.com`

**Yanping Huang**
Google Research
Mountain View, CA 94043
`huangyp@google.com`

**Thang Luong**
Google Research
Mountain View, CA 94043
`thangluong@google.com`

**Henrik Kretzschmar**
Waymo LLC
Mountain View, CA 94043
`kretzschmar@waymo.com`

**Yuning Chai**
Waymo LLC
Mountain View, CA 94043
`chaiy@waymo.com`

**Dragomir Anguelov**
Waymo LLC
Mountain View, CA 94043
`dragomir@waymo.com`

## Abstract

The vast majority of deep models use multiple gradient signals, typically corresponding to a sum of multiple loss terms, to update a shared set of trainable weights. However, these multiple updates can impede optimal training by pulling the model in conflicting directions. We present Gradient Sign Dropout (GradDrop), a probabilistic masking procedure which samples gradients at an activation layer based on their level of consistency. GradDrop is implemented as a simple deep layer that can be used in any deep net and synergizes with other gradient balancing approaches. We show that GradDrop outperforms the state-of-the-art multiloss methods within traditional multitask and transfer learning settings, and we discuss how GradDrop reveals links between optimal multiloss training and gradient stochasticity.

## 1 Introduction

Deep neural networks have fueled many recent advances in the state-of-the-art for high-dimensional nonlinear problems. However, when distilled down to its most basic elements, deep learning relies on the humble *gradient* as the optimization signal which drives its complex algorithmic machinery. Indeed, the desire to properly leverage gradients has spurred a wealth of research into optimization strategies which has led to faster, more stable model training [36].

However, the literature has habitually glossed over an increasingly crucial detail: most gradient signals are sums of many smaller gradient signals, often corresponding to multiple losses. A broad array of models fall under this category, including ones not traditionally considered multitask; for example, multiclass classifiers can be split into a loss per class, and object detectors conventionally break down their predictions along various bounding box dimensions. It is uncertain, and in fact unlikely, that a naïve sum of these individual signals would produce the best solution.

Deep learning theory tells us that the local minima found in single-task models through simple gradient updates are generally of high quality [4]. However, such a claim should be reevaluated in the context of multitask loss surfaces, where minima of each constituent loss may exist at different

network weight settings, which results in many poor minima of the sum loss. Such undesirable minima are avoided if we encourage the network to seek out critical points that are *joint minima* – i.e. critical points that lie near a local minimum of all the constituent loss functions.

To generally address such issues, *deep multitask learning* studies properties of models with multiple outputs and has given birth to methods to balance relative gradient magnitudes [3; 17] or tune the full gradient tensor [38]. Still, methods that explicitly tackle joint loss optimization are rare. Works such as [37; 47] do so by finding a common gradient descent direction for all losses, but such methods operate by *removing* suboptimal gradient components. Such reductive processes are still susceptible to local minima and discourage inter-task competition – competition which evidence suggests can be beneficial [6; 46]. Our proposed method not only provides theoretical guarantees of joint loss minima but also allows gradients to compete, and thus avoids the same pitfalls as reductive gradient algorithms. To the best of our knowledge our method is the first with this set of desirable properties.

We motivate our method, Gradient Sign Dropout (GradDrop), by noting that when multiple gradient values try to update the same scalar within a deep network, conflicts arise through differences *in sign* between the gradient values. Following these gradients blindly leads to gradient tug-of-wars and to critical points where constituent gradients can still be large (and thus some tasks perform poorly).

To alleviate this issue, we demand that all gradient updates are *pure in sign* at every update position. Given a list of (possibly) conflicting gradient values, we algorithmically select one sign (positive or negative) based on the distribution of gradient values, and mask out all gradient values of the opposite sign. A basic schematic of the method is presented in Figure 1.

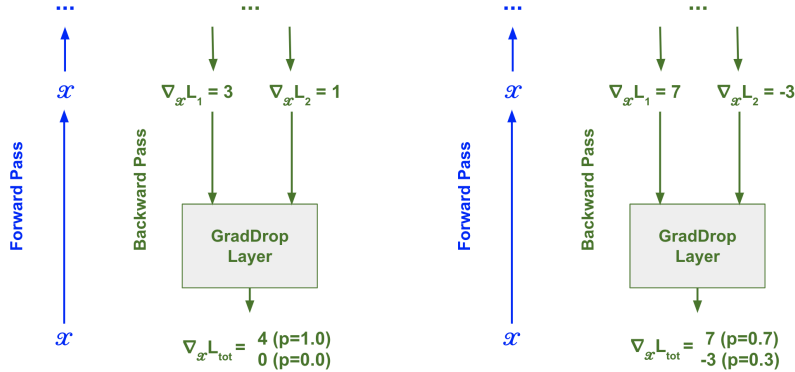

Figure 1: GradDrop schematic for two losses and one scalar. In both cases, we calculate $\mathcal{P}$ (from Equation 1), which tells us the probability of keeping $\nabla$s with positive signs. On the left, $\mathcal{P} = 0.5 * (1 + (3+1)/(|3| + |1|)) = 1.0$, so we keep positive $\nabla$s with 100% probability. On the right, $\mathcal{P} = 0.5 * (1 + (7-3)/(|7| + |-3|)) = 0.7$, so we keep positive $\nabla$s with 70% probability.

The motivation behind GradDrop parallels the well-known relationship between gradient stochasticity and model robustness [18; 39; 40]. When a network finds a narrow, low-quality minimum, the inherent noise within the batched gradient updates serves to kick the model into broader, more robust minima. Similarly, GradDrop assigns a quality score to each gradient update based on its sign consistency, and adds stochasticity along axes where gradients tend to conflict more. An important consequence of this logic is that GradDrop continues triggering until the model finds a minimum that is a joint minimum for all losses (see Section 4.1 for proof).

Our primary contributions are as follows:

1. We present Gradient Sign Dropout (GradDrop), a modular layer that works in any network with multiple gradient signals and incurs no additional compute at inference.

2. We show theoretically and in simulation that GradDrop leads to more stable convergence points than naïve gradient descent algorithms.

3. We demonstrate the efficacy of GradDrop on multitask learning, transfer learning, and complex single-task models like 3D object detectors for a variety of network architectures.

## 2 Related Work

**Optimization via gradient descent** is one of the key pillars of deep learning. Apart from the traditional optimization methods [8; 19; 32; 33; 49], there has been a research thrust on developing different ways to apply gradients to deep networks [2; 7; 10; 15; 36; 45; 50]. The success of such methods comes in part because optimization in single-task models generally converges to high-quality minima [4]. Also important is the relationship between stochasticity and model robustness; as with GradDrop, noisy gradients help repel poor local minima in favor of wider, more robust critical points [18; 39; 40]. These insights are crucial and worth revisiting for multitask environments.

**Multitask learning** presents a challenging problem for optimization, as the loss surface now consists of many smaller loss surfaces. As a subject of study, multitask learning predates deep learning [1; 6], but its power in helping model generalization and transferring information between correlated tasks [30; 48] make it especially relevant in the deep learning era. Although a large part of multitask research focuses on developing new network architectures [16; 20; 23; 25; 28; 29; 31] or new loss functions [17], we focus on methods that explicitly interact with the gradients, which tend to be more lightweight and modular. GradNorm [3] modifies gradient magnitudes to ensure that tasks train at approximately the same rate. MGDA, the Multiple Gradient Descent Algorithm [6; 37], finds a linear combination of gradients that reduces every loss function simultaneously. PCGrad [47] projects conflicting gradients to each other, which achieves a similar simultaneous descent effect as MGDA.

**Many other applications** which are not traditionally considered multitask can benefit from this work. Vision applications such as object detection [24; 34; 35; 51] and instance segmentation [11] explicitly construct multiple losses to arrive at one consolidated result. Language models that employ seq2seq predictions [44] make multiple predictions and create multiple gradient conflicts when backpropagating through time. Domain adaptation and transfer learning [9; 12; 43], topics in which many powerful specialized techniques have been developed, still often rely on multiple losses and thus can benefit from general multitask approaches. Our approach here, although wrapped in the language of multitask learning, has a much wider range of applicability on deep models in general.

## 3 Gradient Dropout

### 3.1 Basic Concepts

Gradient Sign Dropout is applied as a layer in any standard network forward pass, usually on the final layer before the prediction head to save on compute overhead and maximize benefits during backpropagation. In this section, we develop the GradDrop formalism. Throughout, $\circ$ denotes elementwise multiplication after any necessary tiling operations (if any) are completed.

To implement GradDrop, we first define the Gradient Positive Sign Purity, $\mathcal{P}$, as

$$\mathcal{P} = \frac{1}{2}\left(1 + \frac{\sum_i \nabla L_i}{\sum_i |\nabla L_i|}\right). \tag{1}$$

$\mathcal{P}$ is bounded by $[0, 1]$. For multiple gradient values $\nabla_a L_i$ at some scalar $a$, we see that $\mathcal{P} = 0$ if $\nabla_a L_i < 0 \,\forall i$, while $\mathcal{P} = 1$ if $\nabla_a L_i > 0 \,\forall i$. Thus, $\mathcal{P}$ is a measure of how many positive gradients are present at any given value. We then form a mask for each gradient $\mathcal{M}_i$ as follows:

$$\mathcal{M}_i = \mathcal{I}[f(\mathcal{P}) > U] \circ \mathcal{I}[\nabla L_i > 0] + \mathcal{I}[f(\mathcal{P}) < U] \circ \mathcal{I}[\nabla L_i < 0] \tag{2}$$

for $\mathcal{I}$ the standard indicator function and $f$ some monotonically increasing function (often just the identity) that maps $[0, 1] \mapsto [0, 1]$ and is odd around $(0.5, 0.5)$. $U$ is a tensor composed of i.i.d $U(0, 1)$ random variables. The $\mathcal{M}_i$ is then used to produce a final gradient $\sum \mathcal{M}_i \nabla L_i$.

A simple example of a GradDrop step is given in Figure 1 for the trivial activation $f(x) = x$.

### 3.2 Extension to Transfer Learning and other Batch-Separated Gradient Signals

A complication arises when different gradients correspond to different examples, e.g. in mixed-batch transfer learning where transfer and source examples connect to separate losses. The different gradients at an activation layer would then not interact, which makes GradDrop the trivial transformation.

We also cannot just blindly add gradients along the batch dimension, as the information present in each gradient is conditional on that gradient's particular inputs. Generally, deep nets consolidate information across a batch by summing gradient contributions at a trainable weight layer. To correctly extend GradDrop to batch-separated gradients, we will do the same.

For a given layer of activations $A$ of shape $(B, F)$, we imagine there exists an additional weight layer $W^{(A)}$ of shape $(F)$ composed of 1.0s, and consider the forward pass $A \mapsto W^{(A)} \circ A$. $W^{(A)}$ is a virtual layer and is not actually allocated memory during training; we only use it to derive meaningful mathematical properties. Namely, we can then calculate the gradient via the chain rule to arrive at

$$\nabla_{W^{(A)}} L_i = \sum_{\text{batch}} (A \circ \nabla_A L_i) \tag{3}$$

where the final sum is taken over the batch dimension[1]. In other words, premultiplying the gradient values by the input allows us to meaningfully sum over the batch dimension to calculate $\mathcal{P}$ and the $\mathcal{M}_i$s. In practice, because we are only interested in $\nabla_{W^{(A)}}$ insofar as it changes the sign content of $\nabla_A$, we will only premultiply by the *sign* of the input.

### 3.3 Full GradDrop Algorithm

The full GradDrop algorithm calculates the sign purity measure $\mathcal{P}$ at every gradient location, and constructs a mask for each gradient signal across $T$ tasks. We specify the details in Algorithm 1.

---

**Algorithm 1** Gradient Sign Dropout Layer (GradDrop Layer)

---

1: **choose** monotonic activation function $f$                        ▷ Usually just $f(p) = p$
2: **choose** input layer of activations $A$                      ▷ Usually the last shared layer
3: **choose** leak parameters $\{\ell_1, \ldots, \ell_n\} \in [0, 1]$         ▷ For pure GradDrop set all to 0
4: **choose** final loss functions $L_1, \ldots, L_n$

5: **function** BACKWARD($A, L_1, \ldots, L_n$)         ▷ returns total gradient after GradDrop layer
6:      **for** $i$ in $\{1, \ldots, n\}$ **do**
7:          **calculate** $G_i = \texttt{sgn}(A) \circ \nabla_A L_i$           ▷ $\texttt{sgn}(A)$ inspired by Equation 3
8:          **if** $G_i$ is batch separated **then**
9:             $G_i \leftarrow \sum_{\texttt{batchdim}} G_i$
10:      **calculate** $\mathcal{P} = \frac{1}{2}\left(1 + \frac{\sum_i G_i}{\sum_i |G_i|}\right)$          ▷ $\mathcal{P}$ has the same shape as $G_1$
11:      **sample** $U$, a tensor with the same shape as $\mathcal{P}$ and $U[i, j, \ldots] \sim \texttt{Uniform}(0, 1)$
12:      **for** $i$ in $\{1, \ldots, n\}$ **do**
13:          **calculate** $\mathcal{M}_i = \mathcal{I}[f(\mathcal{P}) > U] \circ \mathcal{I}[G_i > 0] + \mathcal{I}[f(\mathcal{P}) < U] \circ \mathcal{I}[G_i < 0]$
14:      **set** newgrad $= \sum_i (\ell_i + (1 - \ell_i) * \mathcal{M}_i) \circ \nabla_A L_i$
15: **return** newgrad

---

For many of our experiments, we renormalize the final gradients so that $||\nabla||_2$ remains constant throughout the GradDrop process. Although not practically required, this ensures that GradDrop does not alter the global learning rate and thus observed benefits result purely from GradDrop masking.

Note also the introduction of the leak parameters $\ell_i$. Setting $\ell_i > 0$ allows some original gradient to leak through, which is useful when losses have different priorities – for example, in transfer learning, we prioritize performance on the transfer set. For more details see Section 4.3.

### 3.4 GradDrop Theoretical Properties

We now present and prove the main theoretical properties for our proposed GradDrop algorithm.

**Proposition 1 (GradDrop stable points are joint minima)**: Given loss functions $L_1, \ldots, L_n$ and any collection of scalars $\mathbf{W}$ for which $\nabla_w L_1, \ldots, \nabla_w L_n$ are well-defined, the GradDrop update signal $\nabla_w^{(GD)}$ at any position $w \in \mathbf{W}$ is always zero if and only if $\nabla_w L_i = 0, \forall i$.

**Proof**: Consider $n$ loss functions, indexed $L_1, \ldots, L_n$, and their gradients $\nabla_w L_i$ for $w \in \mathbf{W}$. Clearly, if $\nabla_w L_i = 0, \forall i$, then that $w$ is trivially a critical point for the sum loss $\sum_i L_i$. However, the converse is also true under GradDrop updates. Namely, if there exists some $j$ for which $\nabla_w L_j \neq 0$, without loss of generality assume that $\nabla_w L_j > 0$. According to Equation 1, $\mathcal{P} > 0$ at $w$. Thus $f(\mathcal{P}) > 0$ (as it is monotonically increasing), so there is a nonzero ($f(\mathcal{P})$) chance that we keep all positive signed gradients and thus a nonzero chance that $\nabla_w^{(GD)} \geq \nabla_w L_j > 0$. $\square$

**Proposition 2 (GradDrop $\nabla$ norms sensitive to *every* loss)**: Given continuous component loss functions $L_i(\mathbf{w})$ with local minima $\mathbf{w}^{(\mathbf{i})}$ and a GradDrop update $\nabla^{(GD)}$, then to second order around each $\mathbf{w}^{(\mathbf{i})}$, $E[|\nabla^{(GD)} L|_2]$ is monotonically increasing w.r.t. $|\mathbf{w} - \mathbf{w}^{(\mathbf{i})}|, \forall i$.

**Proof**: Set $\boldsymbol{\delta} := d\boldsymbol{\delta}_0$ for $|\boldsymbol{\delta}_0| = 1$. To second order, around a minimum value $\mathbf{w}^{(\mathbf{i})}$ a loss function has the form $L_i(\mathbf{w}^{(\mathbf{i})} + \boldsymbol{\delta}) \approx L_i(w^{(i)}) + \frac{1}{2}\boldsymbol{\delta}^T H^{(L_i)}(w^{(i)})\boldsymbol{\delta} = L_i(w^{(i)}) + \frac{1}{2}d^2\boldsymbol{\delta}_0^T H^{(L_i)}(w^{(i)})\boldsymbol{\delta}_0$ for positive definite Hessian $H^{(L_i)}$. Because $\boldsymbol{\delta}_0^T H^{(L_i)}(w^{(i)})\boldsymbol{\delta}_0 > 0$, $\nabla L_i$ at $\mathbf{w}^{(\mathbf{i})} + \boldsymbol{\delta}$ is proportional to $d$. As $d$ increases, so will the magnitude of each $\nabla L_i$ component, which then immediately increases the total expected gradient magnitude induced by GradDrop. $\square$

From Proposition 1, we see that GradDrop will result in a zero gradient update only when the system finds a perfect joint minimum between all component losses. Not only that, but Proposition 2 implies that GradDrop induces proportionally larger gradient updates with distance from **any** component loss function minimum, regardless of the value of the total loss. The error signals induced by GradDrop are thus sensitive to **every task**, rather only relying on a sum signal. This sensitivity also increases monotonically with distance from any close local minimum for any component task. Thus, GradDrop optimization will seek out joint minima, but even when such minima do not strictly exist Proposition 2 shows GradDrop will seek out system states that are at least close to joint minima. For a clear illustration of this effect in one dimension, please refer to Section 4.1.

A potential concern could be that by being sensitive to every loss function, GradDrop updates are too noisy and the overall system trains more slowly. However, that is not the case, as GradDrop updates *on expectation* are equivalent with standard SGD updates.

**Proposition 3 (Statistical Properties)**: Suppose for 1D loss function $L = \sum_i L_i(w)$ an SGD gradient update with learning rate $\lambda$ changes total loss by the linear estimate $\Delta L^{(SGD)} = -\lambda|\nabla L|^2 \leq 0$. For GradDrop with activation function (see Eq. 2) $f(p) = k(p - 0.5) + 0.5$ for $k \in [0, 1]$ (with default setting is $k = 1$), we have:

1. For $k = 1$, $\Delta L^{(SGD)} = E[\Delta L^{(GD)}]$
2. $E[\Delta L^{(GD)}] \leq 0$ and has magnitude monotonically increasing with $k$.
3. $\mathrm{Var}[\Delta L^{(GD)}]$ is monotonically decreasing with respect to $k$.

We present the proof of this proposition in the Appendix, along with generalizing it to arbitrary activation functions. $\square$

Importantly, even though GradDrop has a stochastic element, it provides the same expected movement in the total loss function as in vanilla SGD. Also important is the hyperparameter $k$, which controls the tradeoff between how much the GradDrop update follows the overall gradient and how much noise GradDrop induces for inconsistent gradients. A smaller value of $k$ implies a larger penalty/noise scale, and a value of $k = 0$ means we randomly choose a sign for every gradient value. We call the $k = 0$ case Random GradDrop and show it generally compares unfavorably to $k > 0$, but our evidence does not preclude a situation where the higher noise in the $k = 0$ case may be desirable. Indeed, in most of our experiments the $k = 0$ Random GradDrop setting still outperforms the baseline.

## 4   Experiments with GradDrop

In this section we present the main experimental results related to GradDrop. All experiments are run on NVIDIA V100 GPU hardware. We will provide relevant hyperparameters within the main text, but we relegate a complete listing of hyperparameters to the Appendix. We also rely exclusively on standard public datasets, and thus move discussion of most dataset properties to the Appendices.

All multitask baselines (including PCGrad, to keep compute overhead tractable) and the GradDrop layer are applied to the final layer before the prediction heads to keep compute overhead tractable. We

primarily compare to other state-of-the-art multitask methods, which include GradNorm [3], MGDA [37], and PCGrad [47]. Descriptions of all these methods were given in Section 2.

For completion, we also compare to Gradient Clipping (e.g. [50]) and Gradient Penalty [10]. Although not strictly multitask methods, these gradient-based methods enjoy wide popularity and will provide evidence that principled single-task methods are not enough to optimize a true multitask model.

## 4.1 A Simple One-Dimensional Example

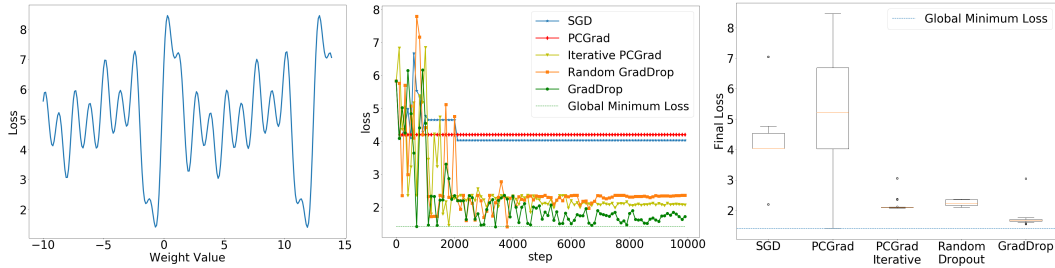

(a) Sum of sinusoids loss function    (b) Loss curves for one random run    (c) Summary results for 200 runs

Figure 2: GradDrop toy example. (a) A synthetic 1D loss function composed of five sines. (b) Loss curves for GradDrop and baselines given a random initialization of the trainable weight. (c) Boxplot of final converged loss values when the methods in b. are run 200 times.

We illustrate GradDrop in one dimension. In Figure 2 we present results on a simple toy system, with a loss function that is the sum of five sines of the form $L(x; a, b) = \sin(ax + b) + 1$. The final loss is shown in Figure 2(a). Note that although each $L_i$ has identical periodic local minima, the sum loss has a wide distribution of local minima of variable quality.

We now initialize the one weight $w$ to a random value and run various optimization techniques for 10000 steps. In Figure 2(b) we plot the loss curves for one example trial. We note that PCGrad [47] does not train in this low-dimensional setting, as any sign conflict would result in PCGrad zeroing the gradients. For fairness, we include a slight modification of PCGrad called iterative PCGrad which still works in low dimensions (for details see Appendix). We also include Random GradDrop, which is a weak version of GradDrop where $f(\mathcal{P})$ is set to $0.5$ everywhere. We see that GradDrop has the best performance of all methods tested. Such a conclusion is further reinforced when we run this experiment 200 times and plot the statistics of the final results, which are shown in Figure 2(c).

Multiple algorithms (GradDrop, Random GradDrop, and Iterative PCGrad) tend to find the deepest minimum, but GradDrop still performs better. We attribute this to the success of our sign purity measure $\mathcal{P}$ at properly emphasizing gradient directions with higher levels of consistency.

## 4.2 Multitask Learning on Celeb-A

We first test GradDrop on the multitask learning dataset CelebA [26], which provides 40 binary attributes based on celebrity facial photos. CelebA allows us to test GradDrop in a truly archetypal multitask setting.

We also use a standard shallow convolutional network to perform this task. Our network consists only of common layers (Conv, Pool, Batchnorm, FC Layers) and contains 9 total layers along with 40 predictive heads. The results of our experiments are summarized in Figure 3 and Table 1.

We see that GradDrop outperforms all other methods. Although the improvements may seem mild in Table 1, they are substantial for this dataset and Figure 3(a) reveals a visually significant effect. Figure 3(b) also shows an ablation study of performance when we choose to marginalize our gradient signal across the batch dimension, as suggested by Section 3.2. Although our gradient signal for CelebA is not batch-separated and thus we are not strictly required to sum the GradDrop signal across our batches, this operation improves GradDrop's memory and compute efficiency, and also can clearly improve model performance. As there are thus few disadvantages from using the sum-over-batch strategy, all further GradDrop runs in this paper will use sum-over-batch.

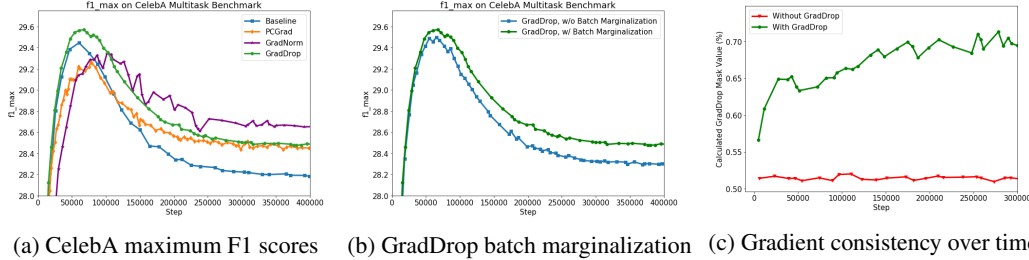

(a) CelebA maximum F1 scores    (b) GradDrop batch marginalization    (c) Gradient consistency over time

Figure 3: Experiments with GradDrop on CelebA.

Table 1: Multitask Learning on CelebA. We repeat training runs and report standard deviations of $\leq 0.04\%$ for F1 Score and $\leq 0.02\%$ for accuracy.

| Method | Error Rate (%) $\downarrow$ | Max F1 Score $\uparrow$ | Speed Compared to Baseline $\uparrow$ |
|---|---|---|---|
| Baseline | 8.71 | 29.35 | 1.00 |
| Gradient Clipping [50] | 8.70 | 29.34 | 1.00 |
| Gradient Penalty [10] | 8.63 | 29.43 | 0.35 |
| MGDA [37] | 10.82 | 26.00 | 0.25 |
| PCGrad [47] | 8.72 | 29.25 | 0.20 |
| GradNorm [3] | 8.68 | 29.32 | 0.41 |
| Random GradDrop | 8.60 | 29.42 | **0.45** |
| GradDrop (ours) | **8.52** | **29.57** | **0.45** |

Furthermore, Figure 3(c) plots the percentage of gradients passed by the GradDrop layer, for both a GradDrop model and a baseline model[2]. This percentage correlates to the degree of sign consistency of gradients at the GradDrop layer. This metric does not improve at all when training the baseline, but improves appreciably when GradDrop is enabled, suggesting that the critical points found by GradDrop have more consistent gradients and thus higher probability of being a joint minimum.

It is interesting to note that GradDrop also overfits less. We posit that GradDrop is a good regularizer due to its tendency to reject weak loss minima that may overfit. The only stronger regularizer may be GradNorm [3], but GradNorm explicitly curtails overfitting with its $\alpha$ hyperparameter.

CelebA with its $T = 40$ tasks also presents us with an excellent opportunity to test method speed. Looking at the last column of Table 1, we see that GradDrop is the fastest of the multitask methods tried (not counting gradient clipping, which is a general single-task method), possibly because it only requires a simple calculation at each tensor position of $\mathcal{O}(T)$ rather than multiple iterative steps like MGDA or $\mathcal{O}(T^2)$ orthogonal projections like PCGrad.

### 4.3 Transfer Learning on CIFAR-100

We now use GradDrop in a transfer learning setting, which is a batch-separated setting (see Section 3.2). We transfer ImageNet2012 [5] to CIFAR-100 [21] by using input batches consisting of half CIFAR-100 and half ImageNet2012 examples. Each dataset has its own predictive head and loss.

We use a more complex network based on DenseNet-100 [13], both to increase performance and to test GradDrop with more complex network topologies. Our results are shown in Table 2 and Figure 4, where we present the best accuracy achieved by each method and the corresponding loss[3]; we include the loss as it is generally smoother.

We see that the best model uses a combination of GradDrop and GradNorm [3], although the GradDrop-Only model also performs well. As in the CelebA experiments presented in Section 4.2, the performance gap is larger when the baseline models overfit later in training. The general synergy between GradDrop and other multitask methods such as GradNorm is important, as it suggests

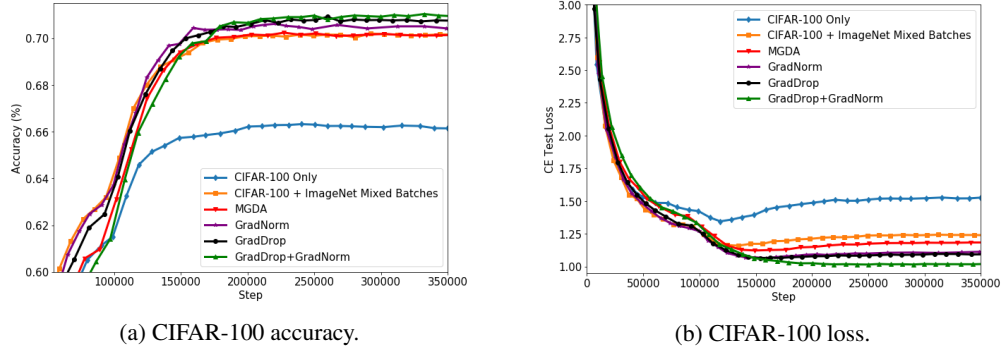

| (a) CIFAR-100 accuracy. | (b) CIFAR-100 loss. |

Figure 4: Accuracy and loss curves for CIFAR-100 transfer learning experiments. In all cases Gradient Dropout outperforms all other methods tried.

Table 2: Transfer Learning from ImageNet2012 to CIFAR-100. We repeat training runs and observe standard deviations of $\leq 0.2\%$ accuracy and $\leq 0.01$ loss.

| Method | Top-1 Error (%) ↓ | Test Loss ↓ |
|---|---|---|
| Train on CIFAR-100 Only | 33.6 | 1.52 |
| Mixed Batch (MB) | 29.8 | 1.22 |
| MB + Gradient Clipping [50] | 29.4 | 1.22 |
| MB + Gradient Penalty [10] | 30.6 | 1.28 |
| MB + MGDA [37] | 29.7 | 1.17 |
| MB + GradNorm [3] | 29.4 | 1.11 |
| MB + GradDrop (ours) | 29.1 | 1.08 |
| MB + GradNorm [3] + Random GradDrop | 29.8 | 1.04 |
| MB + GradNorm [3] + GradDrop (ours) | **28.9** | **1.01** |

GradNorm can add to complex models which already employ an array of pre-existing deep learning tools. We explore this synergy further in Section 4.5.

For our final GradDrop model we use a leak parameter $\ell_i$ set to 1.0 for the *source set*. In this setting, source set gradients are allowed to flow unimpeded but transfer set gradients are masked. This setting is optimal as the source dataset is usually larger and the masking effectively curtails overfitting on the transfer dataset. For more experiments related to the leak parameter, see Section A.5.

### 4.4 3D Point Cloud Detection on Waymo Open Dataset

We now present results on a much more complex problem: 3D vehicle detection from point clouds on the Waymo Open Dataset [42]. For this task we use a PointPillar model [22], a complex and competitive 3D detection architecture that voxelizes a point cloud and uses standard 2D convolutions to derive deep predictive features. We also note that object detection is traditionally considered a *single-task* problem, but still has multiple losses – 3 for each coordinate of the box centers, 3 for each dimension of the box, 1 on box orientation, and (in our formulation) 2 classifiers for box motion direction and box class. Our results thus show that GradDrop is applicable in a much wider context than the traditional explicit interpretation of "multitask learning" might imply.

Our main results are shown in Table 3, where we show Average Precision (AP) and Average Precision w/ Heading (APH) scores (for training curves see Appendix). APH is a metric introduced in [42], which penalizes boxes for being $180^o$ mis-oriented. All runs include gradient clipping at norm 1.0, and we are unable to compare to gradient penalty due to memory restrictions. GradDrop results in marked improvements, especially in the APH metrics. We also note that like the gradient norm methods (which focus on the overall magnitude of gradients rather than their high-dimensional content), GradDrop provides a moderate boost in 2D performance. However, GradDrop does not suffer from the same substantial regressions in 3D performance, and instead improves all metrics across the board.

Table 3: Object Detection from Point Clouds on the Waymo Open Dataset. We report standard deviations of $\leq 0.3\%$ on AP values and $\leq 0.5\%$ on APH values.

| Method | 2D AP (%) ↑ | 2D APH (%) ↑ | 3D AP (%)↑ | 3D APH (%) ↑ |
|---|---|---|---|---|
| Baseline | 76.2 | 69.9 | 57.1 | 53 |
| **Gradient Norm Methods** | | | | |
| MGDA [37] | **76.8** | 69.5 | 20.0 | 18.3 |
| GradNorm [3] | **76.9** | 71.7 | 51.0 | 48.2 |
| **Full Gradient Tensor Methods** | | | | |
| PCGrad [47] | 76.2 | 70.2 | 58.4 | 54.4 |
| Random GradDrop | 76.4 | 66.6 | 57.6 | 50.5 |
| GradDrop (Ours) | **76.8** | **72.4** | **58.8** | **56.0** |

Table 4: Synergy Between GradDrop and GradNorm

| | CelebA | | Waymo Open Dataset | |
|---|---|---|---|---|
| Method | Err Rate (%) ↓ | $F1_{max}$ ↑ | 3D AP (%) ↑ | 3D APH (%) ↑ |
| GradNorm Only | 8.68 | 29.32 | 51.0 | 48.2 |
| GradNorm + GradDrop (ours) | **8.57** | **29.50** | **55.1** | **51.5** |

## 4.5 Synergy with Gradient Normalization and Other Methods

One important property of GradDrop is that it primarily modifies the gradient tensor direction, which is then largely left alone by other deep learning techniques. In principle, GradDrop can thus be applied in parallel with other multitask methods. In this section, we demonstrate positive interactions between GradDrop an GradNorm [3], evidence that GradDrop can be considered a modular part of a diverse toolset which can be applied in a wide array of applications.

Our main results regarding synergy between GradDrop and GradNorm are summarized in Table 4. Along with the CIFAR-100 results in Section 4.3, we find GradDrop often leads to significant improvements when applied with GradNorm. This is especially true where GradNorm performs poorly; for example, although GradNorm tends to regress in the 3D AP metrics compared to baseline, GradDrop+GradNorm recovers much of that performance while still performing well in the 2D AP metrics (see Appendix for 2D AP numbers). We also experimented with GradDrop+MGDA, but with limited success. We hypothesize that MGDA works best when input tensors have explicitly conflicting signs, while GradDrop's final gradient tensors have the same sign (or zero) at all positions.

From an efficiency standpoint, applying GradDrop on top of GradNorm or MGDA comes essentially for free; both GradNorm and MGDA already require us to calculate $\nabla_{\mathbf{W}} L_i, \forall i$, which is the most expensive step in GradDrop. And because we know GradDrop is faster than the other methods described (see Table 1), the additional compute to add GradDrop is small.

## 5 Conclusions

We have presented Gradient Sign Dropout (GradDrop), a method that turns additive gradient signals into a sum signal that is pure in sign and encourages the network to seek out joint minima. From a theoretical standpoint, GradDrop provides superior behavior in the face of suboptimal local minima, and also works for a wide array of network architectures and multitask learning settings.

Apart from our concrete contributions, we also hope that GradDrop will invigorate discussion regarding how best to optimize the complex loss surfaces induced by multitask learning. Our results suggest that the traditional faith in standard gradient descent methods may not describe the full picture, and a realignment of our understanding of optimization robustness to include multitask concepts and gradient stochasticity is prudent as models become ever more complex. We present GradDrop as a crucial early piece of this increasingly important puzzle.

# 6 Broader Impacts

In this paper we presented GradDrop, a general algorithm that can be used as a modular addition to multitask models. At its core, our contribution is the development of a general machine learning algorithm without any assumptions of specific applications, so the potential broader impacts of our work is dependent on the application area.

However, it is also true that multitask learning operates by attempting to leverage multiple sources of potentially disparate information and making joint predictions based on those sources. When applied correctly, multitask models can be less prone to bias/unfairness as they have access to a larger, more diverse source of information. However, when applied incorrectly, multitask models may end up reinforcing the same biases that we want to eliminate; imagine, for example, multitask models which make predictions separately for different subpopulations of the input dataset and due to lack of proper training dynamics end up overfitting to each in turn. Our proposed algorithm may have beneficial effects in combating such overfitting, as our algorithm is effective at finding joint solutions that consistently take all available information into account. As such, we believe that GradDrop will have a positive broader impact on machine learning work by providing ways to arrive at better regularized solutions that are more reflective of reality.

# 7 Funding Disclosure

All funding and resources used to complete the work described in this paper were provided by the employers of the authors, namely Waymo LLC and Google LLC. No third-party or competing sources of funding or resources were used.

## Footnotes

[1]The initialization of the virtual layer is not only meant to keep the forward logic trivial. It is relevant also in the derivation of Equation 3, as it gives us that $\nabla_A L_i = W^{(A)} \circ \nabla_{W^{(A)} \circ A} L_i = \nabla_{W^{(A)} \circ A} L_i$

[2]For the baseline model, this statistic is hypothetical and no gradients are actually masked.

[3]This is the loss that corresponds to the highest accuracy model, not the model with the lowest loss. However, reporting the latter would not change the trend.

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
