[Supplementary Material]

# A   Appendix

The majority of the appendix is devoted to a faithful listing of hyperparameters, datasets, and training settings for all of our experiments. However, we also expand on some intuitions behind our treatment of batch-separated gradients in Section A.2 and present some more experiments on CIFAR-100 transfer learning in Section A.5 and on the Waymo Open Dataset in Section A.6.

## A.1   Addendum on Proposition 3 and Choice of Activation Function

We begin with a proof of Proposition 3, which we rewrite here for convenience:

**Proposition 3:** Suppose for 1D loss function $L = \sum_i L_i(w)$ an SGD gradient update with learning rate $\lambda$ changes total loss by the linear estimate $\Delta L^{(SGD)} = -\lambda |\nabla L|^2 \leq 0$. For GradDrop with activation function $f(p) = k(p - 0.5) + 0.5$ for $k \in [0, 1]$ (with default setting is $k = 1$), we have:

1. For $k = 1$, $\Delta L^{(SGD)} = E[\Delta L^{(GD)}]$
2. $E[\Delta L^{(GD)}] \leq 0$ and has magnitude monotonically increasing with $k$.
3. $\mathrm{Var}[\Delta L^{(GD)}]$ is monotonically decreasing with respect to $k$.

**Proof:** For simplicity of notation and without loss of generality, let us assume a learning rate of $\lambda = 1$. Define $p := \sum_{\nabla_i \geq 0} |\nabla_i|$ and $n := \sum_{\nabla_i < 0} |\nabla_i|$ to be the total absolute value of positive and negative gradients, respectively. From the definition of $\mathcal{P}$ as in Eq. 1 in the main paper, we can easily derive that $\mathcal{P} = p/(p + n)$.

We then calculate

$$f(\mathcal{P}) = k(\mathcal{P} - 0.5) + 0.5 = 0.5\left(\frac{p - n}{p + n}\right)k + 0.5 \tag{4}$$

$$1 - f(\mathcal{P}) = -0.5\left(\frac{p - n}{p + n}\right)k + 0.5 \tag{5}$$

We then note that with total gradient $p - n$, the value $\Delta L$ under GradDrop is precisely

$$E[\Delta L^{(GD)}] = -(p - n)(f(\mathcal{P})p + (1 - \mathcal{P})(-n)) \tag{6}$$

$$= -(p - n)\left(0.5\left(\frac{p - n}{p + n}\right)kp + 0.5\left(\frac{p - n}{p + n}\right)kn + 0.5p - 0.5n\right) \tag{7}$$

$$= -0.5(p - n)\left(\left(\frac{k}{p + n}\right)((p - n)p + (p - n)n) + (p - n)\right) \tag{8}$$

$$= -0.5(p - n)\left(k(p - n) + (p - n)\right) \tag{9}$$

$$= -0.5(k + 1)(p - n)^2 \tag{10}$$

We note that for $k = 1$, this reduces to $-(p - n)^2$, which is precisely $\Delta L^{(SGD)}$, proving the first claim. We also note that the magnitude of this expression is monotonically increasing with $k$, but it is always negative assuming $k \geq 0$, thus proving the second claim.

As for the variance claim, it is straightforward to calculate:

$$\mathrm{Var}[\Delta L^{(GD)}] = E[(\Delta L^{(GD)})^2] - (E[(\Delta L^{(GD)}])^2 \tag{11}$$

$$= (f(\mathcal{P})p^2 + (1 - f(\mathcal{P}))n^2)(p - n)^2 - (0.5(k + 1)(p - n)^2)^2 \tag{12}$$

$$= 0.5(p - n)^2\left(\left(\frac{p - n}{p + n}\right)p^2k - \left(\frac{p - n}{p + n}\right)n^2k + p^2 + n^2\right) - (0.5(k + 1)(p - n)^2)^2 \tag{13}$$

$$= 0.5(p-n)^2 \left( \left( \frac{p-n}{p+n} \right)(p^2 - n^2)k + p^2 + n^2 - 0.5(k+1)^2(p-n)^2 \right) \qquad (14)$$

$$= 0.5(p-n)^2 \left( (p-n)^2 k + p^2 + n^2 - 0.5(k+1)^2(p-n)^2 \right) \qquad (15)$$

$$= 0.5(p-n)^2 \left( (p-n)^2(k - 0.5(k+1)^2) + p^2 + n^2 \right) \qquad (16)$$

$$= 0.25(p-n)^2 \left( (p-n)^2(-k^2 - 1) + 2p^2 + 2n^2 \right) \qquad (17)$$

Although not as simple as our expression for expected value, the variance expression treated as a function of $k$ looks like $A(-k^2 - 1) + B$, with $A, B \geq 0$ and is thus clearly a monotonically decreasing function of $k$ for $k \in [0, 1]$. The third claim is proven. $\square$

Although Proposition 3 was proven for a specific family of activation functions (i.e. $f(p) = k(p - 0.5) + 0.5$), it easily extends to the result that any choice of $f$ that is (1) odd around the point $(0.5, 0.5)$, (2) monotonically increasing, and (3) bounded by $0.0 \leq f(p) \leq 1.0$ will have similar characteristics. Namely, the *steeper* (formal definition to follow) that $f$ is, the higher its corresponding magnitude of $E[\Delta L^{(GD)}]$ and the lower its variance. Namely,

**Corollary 3.1**: Take the family of real-valued continuous activation functions $\mathcal{F}$ such that $f \in \mathcal{F}$ if $f$ is defined on the domain $[0, 1]$, odd around $(0.5, 0.5)$, monotonically increasing, and has output bounded by $0 \leq f(p) \leq 1$ on its domain. We say $f \in \mathcal{F}$ is steeper than $g$ if $f(p) \geq g(p)$ when $p \geq 0.5$ and $f(p) \leq g(p)$ otherwise. For $f, g \in \mathcal{F}$, if $f$ is steeper than $g$, call the corresponding expected loss changes as $E[\Delta L^{(f)}]$ and $E[\Delta L^{(g)}]$. Then the following must be true:

1. $E[\Delta L^{(f)}] \leq E[\Delta L^{(g)}] \leq 0$.
2. $\text{Var}[\Delta L^{(f)}] \leq \text{Var}[\Delta L^{(g)}]$.

**Proof**: It is important to note that the proof for Proposition 3 is true for all values of $p \geq 0$ and $n \geq 0$. That is, the proof for Proposition 3 immediately implies that given any triplet of values $(p, n, \mathcal{P})$, the claims of the proposition are true as a function of $k$. For $\mathcal{P} = 1$, tuning the value of $k$ allows us to sweep the value of $f(1)$ smoothly from 0.5 to 1, and the corresponding value of $f(-1)$ smoothly from 0.5 to 0. Thus, at these two special points, we have access to the full range of possible outcomes. And so if we limit ourselves to the $\mathcal{P} = 1$ and $\mathcal{P} = 0$ cases, we immediately conclude the following:

Given any value of $(p, n)$ and the resultant value of $\mathcal{P}$, if $f(\mathcal{P}) \geq g(\mathcal{P})$ and $\mathcal{P} \geq 0.5$, or if $f(\mathcal{P}) \leq g(\mathcal{P})$ and $\mathcal{P} \leq 0.5$, then $E[\Delta L^{(f)}] \leq E[\Delta L^{(g)}]$ and $\text{Var}[\Delta L^{(f)}] \leq \text{Var}[\Delta L^{(g)}]$ as a special case of Proposition 3.

Because the conditions so listed cover every value for every possible valid activation function $f$ and $g$, the corollary is proven. We also note that $E[\Delta L^{(f)}] \leq 0$ for any $f \in \mathcal{F}$ because the "least steep" activation function is $f(p) = 0.5$, which we showed in Proposition 3 has an expected $\Delta L^{(f)}$ value of $\leq 0$. $\square$

We note that the variance claims in both Proposition 3 and Corollary 3.1 are relatively simple extensions of the intuitive result that the variance of a random variable that can take on only two values is maximized when the two values each have a probability weight of $50\%$. We also note that because of the corollary, the results in Proposition 3 are in fact valid for the extended class of activation functions $f(p) = \text{clip}(k(p - 0.5) + 0.5, 0.0, 1.0)$ for all $k \geq 0$.

### A.2   More Intuition Regarding Batch-Separated Gradients

Perhaps one of the most subtle components of the proposed GradDrop method is its treatment of batch-separated gradients. Although the treatment in the main paper is more mathematical, we would like to use this section to develop some more intuition for our proposed methodology.

As described in Section 3.2, it is necessary to develop a version of GradDrop that operates nontrivially when gradients are incident on orthogonal sub-batches, like in our transfer learning experiments in Section 4.3. The issue we need to resolve is that these gradients are dependent on their batch's input values, so just summing gradients across the batch dimension is not an option. For example,

a gradient value of 4.0 when the input value is 1.0 is not in general the same scenario as a gradient value of 4.0 when the input value is -1.0.

An important insight is that most operations in a standard deep network are *multiplicative* in nature. Although additions of a bias are also standard in neural networks, they are vastly outnumbered by the amount of multiplicative operations and often are left out entirely of the network. However, if our basic building block within a deep network is multiplication, this means that the important quantity is not the pure value of a gradient, but whether that gradient pulls an input value *further or closer to zero*. Thus, the important value when comparing gradients is (input)×(grad), rather than the naked gradient.

However, an additional complication arises because the input is often high variance, and taking this product as our key metric can produce unstable results. An additional modification can be made based on the reasoning that GradDrop operates mainly by reasoning about the sign content of the gradients. The reason why pre-multiplication by the input value is useful is only because it ensures we do not make a sign error when summing multiple gradients together. In that sense, it is sufficient to premultiply by the *sign* of the input, as this allows us to correct our gradient signal for any potential sign errors without being susceptible to the added variance of the inputs.

In the main paper Section 3.2, we derived the proposed rule by assuming a virtual layer that was simple element-wise multiplication at each activation position. In principle, there are also other layers with trainable weights (e.g. dense layers, conv layers) for which we could consider a virtual layer and derive a rule for marginalization of the gradient signal across batches. It is a potential direction of future work to see if any of these other layers result in more robust rules for gradient comparison.

### A.3   A Simple One-Dimensional Example: Addendum

Because the experiment in Section 4.1 uses a model with only one trainable weight, there isn't much to list in terms of hyperparameters. We train all runs with an initial learning rate of 0.2 and a decay ratio of 0.5 applied every 1k steps. Every run is 10k steps in total. We use a standard SGD optimizer.

The sine curves we use to generate the final loss are of the form $\sin(ax + b) + 1.0$. The 1.0 affine factor is only there so that all loss values are nonnegative, which is purely cosmetic. The five sine functions have the following parameters for $(a, b)$:

$$(1.0, 0.0)$$
$$(1.5, 0.2)$$
$$(2.0, 0.4)$$
$$(2.5, 0.6)$$
$$(5.0, 0.8)$$

The sine function periods are selected purely at random and are not chosen to necessarily emphasize any particular behavior.

We note that many methods, such as MGDA [37] and PCGrad [47] do not operate well in the low-dimensional regime. Although it is difficult to adapt MGDA to lower dimensions, we were able to modify PCGrad to exhibit nontrivial behavior in low dimensions. Namely, PCGrad first makes a static copy of the original gradients and then orthogonally projects gradient tensors to each other with reference to the static copy. Instead, we do not make a copy of the original gradient vector and instead update the input gradients in-place. Such a replacement strategy, which we call Iterative PCGrad, adds noise to the PCGrad method but allows for reasonable operation in low dimensions. We also have tried Iterative PCGrad on some of the other experimental settings within this work and it generally seems to perform similarly to PCGrad proper.

### A.4   Multitask Learning on Celeb-A: Addendum

For these experiments we use the Celeb-A dataset in its standard setting. We use the standard 160k/20k dataset split and treat each attribute as a separate task that is trained with a standard binary sigmoid classification loss.

Our network is a shallow convolutional network with nine layers (not counting the maxpool layers or predictive head). With the notation CONV-$F$-$C$ for a convolutional layer of filter size $F$ and number of channels $C$, MAX denoting a maxpool

layer of filter size and stride 2, and DENSE-H a dense layer with $H$ outputs, the layer stack is [CONV-3-64][MAX][CONV-3-128][CONV-3-128][MAX][CONV-3-256][CONV-3-256][MAX][CONV-3-512][CONV-3-512][DENSE-512][DENSE-512][DENSE-40]. GradDrop and other baselines are applied after the final CONV layer. All layers use Batch Normalization [14] except for the final predictive head.

We use an Adam optimizer with $(\beta_1, \beta_2)$ = (0.9, 0.999). Our batch size is 8 and we start with a learning rate of 1e-3, with an annealing rate of 0.96 applied every 2400 steps. All baselines are trained with this set of hyperparameters, with the exception of MGDA [37] for which we had to lower the learning rate by a factor of 100x (otherwise the performance of MGDA was very poor). We train all networks past convergence, but report results from the performance peak. We do this so we also can see the behavior of the system when the model degradation from overfitting is most pronounced.

### A.5 Transfer Learning on CIFAR-100: Addendum

We use CIFAR-100 in its standard setting with a 40k/10k data split. All images (including the ImageNet2012 images) are resized to 32x32 before being input into the network to match the CIFAR-100 image resolution. Image values are divided by 256.0 in preprocessing so that values input to the network lie between 0 and 1. This initial normalization improves training stability especially at the beginning of training.

Our network is based on a DenseNet-100-BC [13] model with $k = 12$. The model has 100 layers in total. We do not use data augmentation to reduce the variance of our training results, and we search for hyperparameters that perform optimally on the transfer learning baseline before applying other baselines with the same set of hyperparameters. We do not use Dropout [41] in our network as it appears to degrade performance. We use batch size 8 each for CIFAR-100 and ImageNet2012 inputs (for a total batch size of 16), with an Adam optimizer with $(\beta_1, \beta_2) = (0.9, 0.999)$ and an initial learning rate of 0.001. The learning rate stays constant until step 100k, at which point it decays by a factor of 0.94 every 2000 steps. We train until convergence, which occurs at around 250k ($\approx$50 epochs). Our best results use the GradDrop activation $f(p) = 0.25(p - 0.5) + 0.5$, showing that this particular system benefits from a higher noise penalty (i.e. see theoretical results in Section A.1).

Both ImageNet2012 and CIFAR-100 inputs share the vast majority of the network, but they are given separate BatchNorm trainable parameters to help alleviate negative effects of the domain shift between the two datasets. We found that training is very unstable in this transfer learning setting without this BatchNorm parameter separation.

Unlike our other experiments, we do not try PCGrad [47]. This is primarily because PCGrad is the trivial transformation when all gradients have nonnegative pairwise dot products. However, because transfer learning produces batch-separated gradient signals (see Section 3.2), all the gradients are already pairwise orthogonal before any additional processing. Thus PCGrad would return the trivial transformation and would perform identically to the baseline.

(a) CIFAR-100 transfer learning with various leak parameter settings.

(b) Final CIFAR-100 transfer learning loss plotted against leak parameter settings.

Figure 5: Experiments with leak parameters on the CIFAR-100 transfer learning setting.

Table 5: Transfer Learning from ImageNet2012 to CIFAR-100 with different leak parameters. Standard deviation values are $\leq 0.2\%$ for accuracy and $\leq 0.01$ for loss.

| $\ell_{source}$ | $\ell_{transfer}$ | Top-1 Error (%) $\downarrow$ | Test Loss $\downarrow$ |
|---|---|---|---|
| 0.0 | 1.0 | 30.5 | 1.23 |
| 0.25 | 0.75 | 30.0 | 1.18 |
| 0.5 | 0.5 | 29.0 | 1.10 |
| 0.75 | 0.25 | 29.1 | 1.06 |
| 1.0 | 0.0 | **28.9** | **1.01** |

In the main paper, we also noted that GradDrop allows for flexibility in leak parameters $\ell_i \in [0.0, 1.0]$, such that the final gradient returned is $\ell_i \nabla + (1 - \ell_i) \nabla^{(\text{graddrop})}$ for a given task $i$. We made the claim that for a transfer learning setting, having the standard $\ell_i = 0, \forall i$ environment is suboptimal as we care more about performance on the transfer task. We further claimed that setting a leak parameter of $\ell = 1.0$ for the *source* dataset while keeping $\ell = 0.0$ for the transfer dataset was the optimal setting for transfer learning.

We present here experiments that empirically justify the above statement. In Table 5 and Figure 5 we show results of a panel of experiments conducted with different leak parameters $\ell_{source}$ and $\ell_{transfer}$. We run the CIFAR-100 experiment with five different settings of $\ell_{source}$ and $\ell_{transfer}$, although for ease of interpretation we keep the sum $\ell_{source} + \ell_{transfer}$ at a constant value of 1.0. We note that there is a clear dependence of performance on the value $\ell_{source} - \ell_{transfer}$. The error values stay generally the same for $\ell_{source}$ close to 1.0, but then rise precipitously, although the same trend manifests as a strong linear dependency in the loss values.

To some, this result may be counterintuitive; if we care more about the transfer set, then it seems reasonable that $\ell_{transfer}$ should be higher and not $\ell_{source}$, to ensure that more transfer gradients are transmitted back through the network. However, we find these results fully consistent with our understanding of GradDrop; as GradDrop primarily filters for consistent gradients, it is optimal to allow the unimportant source set to fully overfit while the transfer set is maximally filtered and regularized. We find that this set of experiments strongly suggests that the effect of GradDrop is beneficial.

### A.6 3D Point Cloud Detection on Waymo Open Dataset: Addendum

We use the Waymo Open Dataset for 3D Vehicle Object Detection also in its standard setting, with a total 1000 segments of 20s 10Hz videos. We split the 1000 segments into the original 798/202 split.

Our re-implementation of Pointpillar [22] is faithful to the topological and threshold hyperparameters of that paper, so we refer the reader to the original work for details. We use 8GPUs and a total batch size of 16, with an Adam optimizer with $(\beta_1, \beta_2) = (0.9, 0.999)$. Our initial learning rate is 0.0015 with a rampup period of 1000 steps. We use a cosine annealing schedule as described in [27] for a total training regime of 1.28 million steps.

[22] describes eight losses for our bounding boxes: three losses for $(x, y, z)$ localization of the box center, three losses for $(h, \ell, w)$ regression of the box dimensions, one loss for the rotational orientation of the box, and one loss for the binary box class (i.e. vehicle or not). In addition to those losses, we add a ninth loss in the form of a directional classifier; we use a standard cross-entropy loss to predict whether a box faces forwards or backwards within the dataset coordinate system. We use this loss as there is an intrinsic ambiguity in the rotation loss that does not penalize a predicted box for being exactly $180^o$ rotated with respect to the ground truth. We find that having this additional directional classifier improves performance dramatically on the APH metrics (which penalize heavily for incorrectly oriented boxes).

For sake of completeness, we show the accuracy curves for our 3D detection experiments in Figure 6. We see that GradDrop produces better accuracy in the 3D detection metrics and this benefit is present throughout most of training. PCGrad [47] also performs well, but falls short of the GradDrop performance. We attribute this differential to the ability of GradDrop to more effectively choose consistent gradient directions, a conclusion also supported by our toy experiments in Section 4.1.

| (a) 3D AP | (b) 3D APH |

Figure 6: 3D AP and 3D APH metrics for Waymo Open Dataset.

As with other experiments the MGDA baseline seems to perform relatively poorly, especially in the 3D metrics. We note that because MGDA seeks the linear combination of gradients that results in the smallest norm, tasks which tend to backpropagate higher gradients will become attenuated by MGDA. GradNorm has a similar effect, but because GradNorm's reference point is the mean norm of all gradients rather than the minimum norm of all possible linear combinations of gradients, the effect is much less acute. Because GradNorm also tends to regress in the 3D metrics, we conclude that the 3D-relevant losses ($z$ localization and box height regression) tend to backpropagate higher gradients, which then has a slight negative interaction with GradNorm and a more severe negative interaction with MGDA.

We also present a more extensive set of results in Table 6 for our experiments with synergy between GradDrop and other multitask learning methods, such as GradNorm [3] and MGDA [37]. In the main paper text we only presented 3D metric results as they were where we saw the most prominent effect, but here we tabulate 2D metrics as well and also present results for MGDA+GradDrop. As described in Section 4.5, the effect of applying GradDrop atop MGDA is murky; we see a regression in the 2D metrics but an improvement in 3D. As also discussed in Section 4.5, we attribute this effect to MGDA not behaving properly when its input gradients have the same sign at every position. However, when applied with GradNorm we see that GradDrop significantly improves the 3D metrics while it does approximately as well if not slightly better in the 2D metrics. This result is important, as GradNorm generally provides a moderate boost already in the 2D metrics over the baseline model. The ability of GradDrop to improve the 3D metrics while maintaining the GradNorm advantage in 2D is encouraging.

Table 6: Object Detection from Point Clouds on the Waymo Open Dataset - Synergy With Other MTL Methods

| Method | 2D AP (%) ↑ | 2D APH (%) ↑ | 3D AP (%) ↑ | 3D APH (%) ↑ |
|---|---|---|---|---|
| MGDA [37] | 76.8 | 69.5 | 20.0 | 18.3 |
| MGDA [37] + GradDrop | 73.7 | 65.0 | 32.3 | 28.6 |
| GradNorm [3] | 76.9 | **71.7** | 51.0 | 48.2 |
| GradNorm + GradDrop [3] | **77.3** | **71.6** | **55.1** | **51.5** |