[Reviews · NeurIPS 2020]

Review 1

Summary and Contributions: This paper focuses on the multiple loss optimisation for the same set of weights. They introduce a probabilistic masking technique on the gradients to encourage the consistent gradient update between different losses at backpropagation. And the proposed method can be applied to both the multitask and transfer learning problems.

Strengths: 1. Using the gradient sign to deal with the gradient conflict in multi-loss optimisation is reasonable and interesting. 2. It is nice to see the authors aim to provide some theoretical analysis of the proposed method. 3. The proposed method can be applied to both multitask and transfer learning problems.

Weaknesses: 1. It is not clear in Section 3.4 why the proposed GradDrop gives better update than without using it. 2. It is doubtful whether the competitors were well trained in Table 1. 3. It is not clearly explained about the setting in Section 4.3, and the performance improvement looks fairly marginal in Table 2. 4. From the experimental evaluation in 4.1, it is surprising that Random GradDrop achieves the very strong result. The authors should put more comparison to this competitor in the rest of the experimental evaluations. This is to verify the effectiveness of the proposed Sign GradDrop more thoroughly.

Correctness: Sounds correct.

Clarity: The clarify is overall ok.

Relation to Prior Work: Reasonably clear.

Reproducibility: Yes

Additional Feedback: Can the authors explain more clearly why MGDA/GradNorm performed badly in Table 3? How about the performance on the NYUv2 benchmark. Update: thanks, the authors, for providing the rebuttal. Some of my concerns have been addressed in the rebuttal. I agree that the proposed GradDrop improves the multitask optimization to some extend. However, it seems GradDrop does not improve over the Random GradDrop with a noticeable margin, which reduces the significance of the proposed method. Thus, I keep my original score 5.


Review 2

Summary and Contributions: This paper proposes a novel method that can be used as a regularizer for multi-task learning, where there are multiple loss gradient signals. A naive way would be to take a sum of those signals, but there may be conflicts between them, especially when signs differ with each other. Based on the distribution of gradient values, the paper proposes an algorithm called GradDrop to select either positive or negative sign, and mask out all gradient values of the other sign. The paper explains that the motivation of GradDrop is to assign a score based on its "sign consistency" and try to force the network to find better robust minima. Synthetic and benchmark experiments demonstrate the characteristics of the GradDrop, and show it is better than baseline methods.

Strengths: - The proposed idea and algorithm of the method is simple and novel, and the intuition/motivation behind it is also clear. - Synthetic experiments demonstrate the characteristics of the proposed method. - It is compatible with other methods, especially GradNorm, which makes the proposed method practical. - Experiments show that w/ GradDrop, the consistency of the gradient increases overtime, which is expected.

Weaknesses: - It would make the experiments stronger if Table 3 and 4 reported for several runs (it seems to be a single run.) - Although there are already some experiments that show GradDrop is behaving as expected (like Fig.3) perhaps a more direct evaluation would be to try some synthetic experiments that controls the gradient conflicts, and see what happens when conflict in increased, and see if the GradDrop will become more superior than baselines. I am curious if having similar signals between tasks will diminish the benefits of GradDrop or not.

Correctness: yes

Clarity: yes

Relation to Prior Work: yes

Reproducibility: Yes

Additional Feedback: - Table 1 and 2 explains that the experiments were repeated for several runs, but I couldn't find the number of runs. Also, since there are space, it would be more informative if st dev information was included for each method. - Are Fig.3 and Fig.4 showing the mean of several runs? ---------------------------- After rebuttal period: Thank you for answering my questions. It is good to hear that the authors are planning to add more results to address the concerns, if accepted.


Review 3

Summary and Contributions: The paper builds upon a clever observation: dropout can be applied to better balance gradients while flowing in the backward pass, for the sake of regulating and balancing the contribution of different losses that can be applied to the very same backbone network. The computational methods that authors devise is simple, intuitive, yet effective: several diverse applications are considered and the overall performance is impressive under both error and generalization metrics.

Strengths: 1. The paper reads well. 2. The idea is extremely intuitive, appealing and simple: I am glad to regisiter that authors seem to capitalize from Occam's razor! 3. The idea is novel: I never saw dropout applied in the backward pass only, usually it is the very way around since being applied in the forward pass (and gradients are updated accordingly). 4. The experimental evaluation is broad and compelling: several applications are considered (multi-task learning, domain adaptation, point cloud detection) and prior relevant methods are outperformed in terms of both error and generalization methods

Weaknesses: 1. Authors forgot to compare with two popular approaches that are used to cope the balancing of different loss functions: gradient clipping GC (as in [Zhang et al. Why gradient clipping accelerates training: A theoretical justification for adaptivity, ICLR 2020]) and gradient penalty GP (as in Wasserstein GANs). I suspect the method proposed by authors to be superior, but I still think that an evaluation against such baseline is extremely important given the resonance that GC and GP have in the community.

Correctness: I did not found any error.

Clarity: Absolutely yes

Relation to Prior Work: Yes. The idea is sharply different from the current practice in DL in which the contribution of different loss functions are simply fused together at the gradient stage.

Reproducibility: Yes

Additional Feedback: FINAL EVALUATION I was extremely pleased while reading the paper: I simply found the idea extremely fresh and effective. I have really no problems with the paper, except to the following point: I was really eager to check to which extent the proposed method could outperform GC and GP, given their common usage among deep learning practitioners. I believe that authors should have compared with this approaches: with this additional experiment, I believe that this paper could enter the top 50% of NeurIPS papers. But, without it I do not feel confident in promoting for a full acceptance at this stage. I really encourage authors in providing such experiments at the rebuttal stage: given this additional experiments I would be more than happy to sharply raise my score. MAIN SUGGESTION: compare with GC and GP. I believe they are fundamental baseline methods to be included! Minor Suggestions: 1. Authors could visualize the probability $\mathcal{P}$ in Fig. 1, this will help the reader in getting the proposed method in a glimpse. 2. Authors are slightly out of the recommended page limit. I am aware of the policies that suggest reviewers to be elastic in being tolerat towards a small break of the rules: this is definitely a minor violation. But, I still think that authors could have easiliy made the paper compliant with the 8 page limits by a \vspace or making one of the Tables smaller. 3. I would encourage authors to share the code for the method (although the approach and the pseudocode are so straight that I feel not impossible to implement the paper from scratch). FINAL EVALUATON (AFTER AUTHORS' FEEDBACK) I was really pleased to see that authors addressed all the comments I provided. The experiments I requested showed in a clear manner the effectiveness of the proposed approach versus gradient clipping and penalty, which are now baseline methods that can be added to the manuscript and whose performance the current proposed method improves. Overall therefore, I want to raise my score towards a full acceptance of manuscript and I wish the paper could be accepted as a full paper to NeurIPS 2020.


Review 4

Summary and Contributions: This paper proposed an optimization regularization method for multitask training called ‘gradient sign dropout’, presenting as an extra layer in network. The layer utilizes signs of one layers’ input activations and their corresponded gradients from different loss functions to formulate a Bernoulli distribution and randomly propagates positive or negative partial derivatives according to it. Theoretical proof of the method’s impact on avoiding nearing-zero gradients caused by different losses is provided. Experiments on simple optimization, multitask learning, transfer learning and 3D object detection are provided.

Strengths: - The random scheme used by gradient sign drop can surely prevent sluggish training confronting small gradients caused by composed gradients from different losses. Also, determination of parameters of probability distribution by Equation 1 is somewhat novel. - Impact on preserving gradient consistency is theoretically proved and verified in experiments. - The method brings performance improvements on various tasks and metrics in experiments.

Weaknesses: - The proposed ‘grad drop’ seems been the main idea of this work, which seems not adequate to be total contributions of a NeurIPS paper. - The previous paper ‘Tseng, et al: Regularizing Meta-Learning via Gradient Dropout, arXivPrePrint, 2004.05859’ proposed similar idea for gradient dropping but mainly for meta-learning and with manually adjust distribution parameters, which reduces novelty. Also, this derives another consideration: Is the determination of probability in Equation 1 generally better than manually selected constants? Ablation study on this part is needed. - The method can surely guarantee the consistency since Bernoulli distribution will certainly not have parameters $p=1-p=0$, which makes proof of proposition 1 somewhat trivial. - ‘The key’ described in L136 and L137 is correct but the extension on ‘stable points under joint minima’ needs more concerns: Is the joint minima always existing in multitask learning? L27 introduced the definition of joint minima but failed to mention the evaluation of ‘near the local minima’. For instance, the combination of function $(x-2)^2$ and $(x-1)^2+3$ has minima point at $x=1.25$ but their own minima respectively lay on $x=1$ and $x=2$. In this condition, is the $x=1.25$ ‘near’ enough to be a joint minima? I call for possible theoretical or empirical explanations. - Graddrop actually selects part of loss functions but not total loss to optimize targets. It preserves consistent gradient to pursue joint minimum but the convergence of this algorithm is not proved. If joint minima do exist in a task, how does Graddrop ensure weights converging at ‘stable points’ but not ‘jumping across’ minima of different loss functions? If theoretical analysis is difficult, experiments on convergence are also acceptable. Figure 3 (c) failed to show conspicuous convergence of GradDrop. - The pre-multiplication for extension to batch-separated gradients in 3.2 is hard to be understood. See details in Correctness below. - SGD and PCGrad are trapped in local minima in experiment of section 4.1 regarding Figure 2 (a) and (b). Would reset the learning rate schedule be helpful for these methods? - From Figure 3 (a), the GradDrop achieves the highest maximum F1 in initial stage of training but all of other methods actually HARM the baseline in this stage. In later period, the GradNorm outperforms GradDrop for the reason explained in L189. These two situations weaken persuasion of GradDrop’s effectiveness and universality. Similar problems occurs in ‘Error Rate’ and “Max F1 Score’ columns in Table 1. - The Err Rate and Max F1 Score on CelebA combining GradNorm and GradDrop benefits the former but actually harms the GradDrop itself (8.52->8.57, 29.57->29.50), comparing Table 1 and 4. The performances on Waymo meets similar problem. Although the synergy certainly improves performances of GradNorm as a modular part, the synergy’s harmful impact on GradDrop itself weakens this property. Also, the limited success of GradDrop+MGDA in L239 and L240 makes the experiments on synergy with ‘Other Methods’ either not persuasive or not enough. - Most of experiments are taken with GradDrop added on the final layer between prediction heads in section 4. Providing experiments results of adding on other layer will be favorable.

Correctness: - The implementation of GradDrop in section 3.1 and Algorithm 1 is rigorous and correct as an applicable algorithm. - The proof of theoretical properties in section 3.4 is correct and its trivial extension is acceptable under certain conditions. - The derivation of pre-multiplication with sign in section 3.2 is hard to be understood. The authors claim the situation ‘the information present in each gradient is conditional on that gradient’s particular inputs’ and their motivation ‘to correctly extend’ from L104 to L107. The derivation below, however, simply takes examples of a virtual layer composed by 1.0s and directly extends the results. However, as is mentioned in footnote under page 4, $\nabla_AL_i=\nabla_{W^{(A)} \circ A}L_i$ simply because the virtual layer is identical transform (all 1.0s). If not, these two gradients will not be the same. I understand the ‘virtual’ here may ease the confuse above but I do not understand the relation between this ‘virtual’ layer’s gradient $\nabla_{W^{(A)}}L_i$ and the actual backproped gradients to former layers. In fact, those gradients is related to $\nabla_AL_i$ which is related to value of $W^{(A)}$ but not directly to value of its gradients $\nabla_{W^{(A)}}L_i$. More reasonable and clear explanation is needed besides description from L112 to L115.

Clarity: - The paper is fluently written with topic emphasized and is easy for me to understand its contributions. - Topic of section 4.5 may need further consideration since lack of experiments with “other methods” but just with GradNorm.

Relation to Prior Work: - The paper clearly discusses its relation to previous works and takes many contrast experiments with them. - The paper ‘Tseng, et al: Regularizing Meta-Learning via Gradient Dropout, arXivPrePrint, 2004.05859’ can probably be a cited work

Reproducibility: Yes

Additional Feedback: The authors could enrich the contents of their contributions and provide more clear derivation and supplement experiments results. See details in Weaknesses. afer rebuttal: thanks for the authors's feedback, I am glad that the authors compare results with related paper Tseng, et al, though it is not compulsary,however, only few of my concerns are answered, I have to hold on my original ratings.

[Author Response · NeurIPS 2020]

We humbly thank all of our reviewers for their time and effort in considering our paper. We are encouraged by the general appreciation of the simplicity and wide applicability of GradDrop in our diverse test scenarios. If given the privilege of proceeding, we will address all comments (including any we couldn't address here) within the final version.

[**R1**] We want to reassure **R1** that our toy example methods are well-trained (and repeated 200 times), but SGD/PCGrad perform poorly due to the 1D setting. E.g. it is well known that SGD performs poorly with local minima, and our 1D setting manifests this drawback in a dramatic way. In regards to Table 2, our improvements lie well outside error bars and milder improvements on CelebA are typical (e.g. see [34] and [41]). We included fewer setting descriptions in Section 4.3 because most details were standard, but please reference Section A.4 for a fuller description. We can also certainly apply GradDrop to NYUv2, but chose the Waymo Open Dataset as it is more modern, larger, and difficult.

[**R2**] Regarding experiment stats in Tables 3/4, we did perform multiple runs, and we will include the error bars in a final version. But we should mention that our method's improvements lie well outside any error bars. We did not include per-method variances as they were similar across methods, but can clarify that within the paper. We also appreciate your suggestion on the synthetic experiments; we can already show that more closely overlapping sines within our toy setting will reduce the performance gap between methods, and will include such a study in a revision.

[**R3**] First, there may have been a small misunderstanding regarding page limits, as the Broader Impacts section is not included in the 8-page limit. But more importantly, new Grad Clipping (GC) and Grad Penalty (GP) baselines are shown in Table I. GradDrop handily beats both methods. During training, GP tends to converge faster, but often converges to a worse value. GP is also $\approx$30% slower per step than GradDrop on CelebA, and $\approx$45% slower on CIFAR-100. For 3D detection, we cannot use GP as it takes up too much memory, but in fact GC *was already in use* in our main paper results. So, GradDrop not only beats GC for 3D detection, but can be applied *together* with GC in synergistic fashion. Indeed, we also tested GC+GradDrop on CIFAR-100 and it beats GC-Only by 0.3% accuracy (not shown in Table I).

[**R4**] Regarding pre-multiplication, the 1.0 initialization is necessary as otherwise the layer nontrivially transforms its inputs and is no longer "virtual." Please also see additional discussion in A.1. Thanks for citing Tseng et al; although this counts as contemporaneous work, we will add discussion in a revision but also note that despite the cosmetic similarity of randomly dropping gradients, their work operates on a specific metalearning setting, assumes a random distribution of added grads, and does not consider the *sign* of the gradient, which is at the heart of multitask gradient conflict. We thus believe GradDrop is significantly different and provides important insight not present elsewhere.

[**R1**, **R4**] Regarding MGDA/GradNorm performance, please see Section A.5. We also agree that combining methods is optimal only sometimes (e.g. CIFAR-100), but just having this option is compelling. As to the crux of GradDrop's efficacy: **Proposition 2:** Given continuous, lower-bounded component loss functions $L_i(\mathbf{w})$ with local minima $\mathbf{w^{(i)}}$ and a GradDrop update $\nabla^{(GD)}$, then to second order around each $\mathbf{w^{(i)}}$, $E[|\nabla^{(GD)}L|_2]$ is monotonically increasing w.r.t. $|\mathbf{w} - \mathbf{w^{(i)}}|, \forall i$. **Sketch of proof:** Expand $L_i$ around $\mathbf{w^{(i)}}$ to second order to show that $|\nabla L_i|$ increases w.r.t. $|\mathbf{w} - \mathbf{w^{(i)}}|$.$\square$ As **R4** mentions, we do not claim convergence properties, but GradDrop produces larger gradients when *any* loss term is far from a minimum. Thus, if a model converges, the convergence point *likely* has better overlap between component loss minima. Similarly, any "minimum hopping" as posited by **R4** will favor overlapping minima.

[**R1**, **R4**] More on why we outperform SGD and Random GradDrop (RGD): **Proposition 3:** Suppose for 1D loss function $L = \sum_i L_i(w)$ an SGD grad reduces total loss by a linear estimate $|\Delta L^{(SGD)}|$. For GradDrop (GD) with prob keep fn $f(x) = x$ and RGD, we have $|\Delta L^{(SGD)}| = E[|\Delta L^{(GD)}|] \geq E[|\Delta L^{(RGD)}|]$. **Proof:** Set $p = \sum_{\nabla_i \geq 0} |\nabla_i|$ and $n = \sum_{\nabla_i < 0} |\nabla_i|$. Then $E[|\Delta L^{(GD)}|] = (p - n)(\mathcal{P}p - (1 - \mathcal{P})n) = (p - n)^2 = \Delta L^{(SGD)} \geq 0.5(p - n)^2 = E[|\Delta L^{(RGD)}|]$.$\square$ Thus, GradDrop preserves SGD statistics on average, but unlike SGD can also detect and penalize inconsistent task grads. We also beat RGD both theoretically (Prop 3) and empirically (Table I and main paper Table 3).

[**R1**, **R2**, **R3**, **R4**] Thank you all again for your feedback. We believe GradDrop is not only general and practical, but also gives new insight into the challenges of multitask optimization that will be of interest to the NeurIPS community.

Table I: New Baselines. Std dev per metric, CelebA: ($\pm 0.02\%$, $\pm 0.04$ and $\pm 0.05$) and CIFAR-100: ($\pm 0.2\%$, $\pm 0.02$)

| | **CelebA** | | | **CIFAR-100** | |
|---|---|---|---|---|---|
| Method | Err Rate (%) $\downarrow$ | F1$_{max}$ $\uparrow$ | Test Loss $\downarrow$ | Err Rate (%) $\downarrow$ | Test Loss $\downarrow$ |
| Baseline | 8.71 | 29.35 | 8.00 | 29.8 | 1.22 |
| Gradient Clipping (GC) **R3** | 8.70 | 29.34 | 7.93 | 29.4 | 1.22 |
| Gradient Penalty (GP) **R3** | 8.63 | 29.43 | 7.96 | 30.6 | 1.28 |
| Random GradDrop (RGD) **R1**, **R4** | 8.60 | 29.42 | 7.86 | 29.6 | 1.16 |
| Ours | **8.52** | **29.57** | **7.80** | **28.9** | **1.06** |

[Meta-Review · NeurIPS 2020]

The reviewers generally agree that this paper presents a simple, intuitive and interesting idea. The nature of the method (being a generally applicable optimization enhancement) means that a very thorough experimental evaluation is desirable to convince the reader. However, the rebuttal helped to alleviate the corresponding concerns, and overall this looks like a paper that the community would appreciate.